# Scanning Probe Lithography: State-of-the-Art and Future Perspectives

**DOI:** 10.3390/mi13020228

**Published:** 2022-01-29

**Authors:** Pengfei Fan, Jian Gao, Hui Mao, Yanquan Geng, Yongda Yan, Yuzhang Wang, Saurav Goel, Xichun Luo

**Affiliations:** 1Centre for Precision Manufacturing, Department of DMEM, University of Strathclyde, Glasgow G1 1XQ, UK; pengfei.fan@strath.ac.uk (P.F.); jian.gao@strath.ac.uk (J.G.); 2Key Laboratory of Biomass Chemical Engineering of Ministry of Education, College of Chemical and Biological Engineering, Zhejiang University, Hangzhou 310027, China; huimao@zju.edu.cn; 3Center for Precision Engineering, Harbin Institute of Technology, Harbin 150001, China; gengyanquan@hit.edu.cn (Y.G.); yanyongda@hit.edu.cn (Y.Y.); wangyuzhang67@163.com (Y.W.); 4School of Engineering, London South Bank University, 103 Borough Road, London SE1 0AA, UK; goels@lsbu.ac.uk; 5University of Petroleum and Energy Studies, Dehradun 248007, India

**Keywords:** nanofabrication, scanning probe lithography (SPL), scanning probe microscopy (SPM), nanostructures

## Abstract

High-throughput and high-accuracy nanofabrication methods are required for the ever-increasing demand for nanoelectronics, high-density data storage devices, nanophotonics, quantum computing, molecular circuitry, and scaffolds in bioengineering used for cell proliferation applications. The scanning probe lithography (SPL) nanofabrication technique is a critical nanofabrication method with great potential to evolve into a disruptive atomic-scale fabrication technology to meet these demands. Through this timely review, we aspire to provide an overview of the SPL fabrication mechanism and the state-the-art research in this area, and detail the applications and characteristics of this technique, including the effects of thermal aspects and chemical aspects, and the influence of electric and magnetic fields in governing the mechanics of the functionalized tip interacting with the substrate during SPL. Alongside this, the review also sheds light on comparing various fabrication capabilities, throughput, and attainable resolution. Finally, the paper alludes to the fact that a majority of the reported literature suggests that SPL has yet to achieve its full commercial potential and is currently largely a laboratory-based nanofabrication technique used for prototyping of nanostructures and nanodevices.

## 1. Introduction

The development of nanotechnology is inextricably linked to the downscaling of nanostructures with feature dimensions below 100 nm [1]. Various nanometric structures, such as nano-dot arrays, nano-grooves, and even three-dimensional (3D) nanostructures [2], have been explored for dense nano-products and electronic devices, nanophotonics [3], and biomedical research [4] using diamond tip-based machining [5,6]. The most generally used methods for rapidly prototyping the nanostructures are electron beam lithography (EBL), focused ion beam fabrication (FIB), nanoimprint lithography (NIL), and scanning probe lithography (SPL). A comparison of these nanofabrication techniques is shown in Table 1.

The biggest advantage of EBL and FIB is that they can pattern nanostructures with feature size in sub-5 nm and can process a wide range of materials, such as metals [8], alloys [15], hard and brittle ceramics and semiconductors [13], and polymers [16], making them extremely suitable techniques to process nanostructures with high precision [17]. However, the processing environment is harsh and requires vacuum operation. FIB is similar to EBL in the limitation of processing speeds. For example, the FIB deposition rate can reach a maximum of up to 0.05 μm^3^/s [13]. The processing speed of EBL is dependent on the type of electron beam resist. For example, the patterning speed of EBL can achieve 17 nm/min [10], 40 nm/min [11], and 58 nm/min [12] against SML resist, poly (GMA-co-MMA-co-TPSMA) resist, and IM-MFP_12-8_ resist, respectively. Furthermore, FIB can be destructive and modify the electrical properties of the substrate surface via undesirable deposition of gallium ions. Additionally, the equipment is expensive, which increases the cost of the nanofabrication process. As opposed to EBL and FIB, NIL is relatively low cost and offers simplicity to fabricate a master mold during the nanoimprint process. However, the mold is inflexible and can only replicate the designed nanostructures, which limits the application of the NIL technique. The SPL nanofabrication technique utilizes scanning probe microscopy (SPM) to carry out nanofabrication with the characteristics of low machinery requirements offering simple use, flexible controlling position capability, a relatively high material removal rate, and non-destructive inspection. Functionalized SPM tips can image and manipulate environments at the atomic and the sub-nanometer scale on the surface of a substrate, creating high temperatures, high electric and magnetic fields, high fluxes of many types, and rapid temporal and spatial variations of all of the above and more. It potentially creates a unique, localized, controllable “manufacturing environment,” wherein new methods for controlled nanofabrication are possible. In this review, we analyze SPM techniques as the primary motivation of this paper. 

This paper will briefly traverse through the history of SPM and then illustrate fabrication mechanisms, research status, and merits and drawbacks of major SPM-based nanofabrication approaches and their applications. Finally, this paper will also discuss and compare major SPL nanofabrication approaches and point out the current challenges and outlook of future research directions of the SPL nanofabrication technique. 

## 2. History of SPM 

The generic scanning probe microscope (SPM) is a branch of microscopy that employs a physical tip to scan the workpiece surface to reveal its topography. SPM is a versatile instrument that has been thriving since its invention in 1981 by Binning and Rohrer, leading them to win the Nobel Prize in 1986 [18]. The invention of SPM not only marked the birth of new technology for imaging and analyzing material surface at the nanoscale, but also triggered an unprecedented innovation for maskless nanofabrication or even close-to-atomic scale fabrication via the two most popular family members of SPMs: scanning tunneling microscope (STM) and atomic force microscope (AFM). 

The precedent of close-to-atomic scale fabrication dates back to 1990. Eigler and Schweizer posited an atomic-scale logo of IBM by manipulating Xe atoms on a Ni workpiece for the first time by employing STM [19]. Thereafter, AFM was first utilized as a powerful machine tool to modify the material surface, such as a polycarbonate surface in 1992 [20] and a gold surface in 1997 [21]. Undergoing nearly 30 years of development, the family of SPMs has expanded rapidly, leading to innovations such as electrostatic force microscopy (EFM) [22], magnetic force microscopy (MFM) [23], fluidic force microscopy (FluidFM) [24], piezoresponse force microscopy (PFM) [25], etc. Consequently, a variety of SPL nanofabrication techniques now exist that can offer atomic manipulation, electric field emission, chemical diffusion, electrochemical reaction, thermal deposition, and mechanical scratching. To date, the SPL nanofabrication technique has been deemed a practical method to implement nanofabrication and close-to-atomic scale fabrication. 

## 3. Major SPL Nanofabrication Approaches

The substrate surface can be modified by a probe in the SPL operation in terms of chemical versus physical or removal versus addition. Considering different driving mechanisms in the SPL nanofabrication process, SPL nanofabrication approaches can be classified into close-to-atomic scale SPL, oxidation SPL, thermal SPL, thermochemical SPL, dip-pen SPL, and bias-induced SPL. 

### 3.1. Close-to-Atomic Scale SPL

#### 3.1.1. Fabrication Mechanisms

SPL can reach close-to-atomic scale precision in scanning tunneling microscope (STM) work mode. When a sharp metal tip with a biased voltage approaches a conductive surface within a 1 nm vacuum gap, a quantum tunneling current (several picoamperes to nanoamperes in magnitude) shows a monotonic exponential variation with tip-surface distance. Subsequently, a resolving downscale to individual atom topography of the workpiece can be mapped. When the distance between tip and workpiece decreases to hundreds of pm continuously, an existing force exerted by the tip can cause the workpiece atom within the adsorption site to hop to an adjacent empty site to complete single-atom machining. Additionally, the measurement of threshold force is realized. Although AFM and STM were invented almost simultaneously, the machining capability of AFM and STM has taken different paths. AFM-based machining has various versatile machining approaches depending on the AFM scanning type, which is far beyond STM-based machining. In terms of the AFM work principle, there are three modes (contact, tapping, and non-contact) in AFM that are categorized in terms of the exiting force type between probe and sample surface. The tapping mode-based machining approach in AFM can carry out atomic manipulation at room temperature. The fundamental cause of the capability of atom manipulation is the atomic-scale resolution. The atomic-scale resolution of AFM relies on the detection and precise quantification of the short-range bonding interaction forces (normally tens of picoNewtons to nanoNewtons) between the headmost atom of the tip and workpiece surface atom that is closest to the tip’s headmost atom. Therefore, in the tapping mode, a highly sensitive and powerful detector that can accurately tune the interaction forces by regulating the oscillation of the cantilever results in atom manipulation. In particular, hydrogen depassivation lithography (HDL) is one of the most successful atomically precise manufacturing methods, which is also based on SPM tips to achieve atomic-scale precision [26]. The hydrogen atoms on the (001) surface of Si could be selectively removed by the precise positional control of the atomically sharp tip. The tunneling current passing from the tip to the sample could break the Si-H bonds and depassivate the silicon surface. The position and degree of desorption can be parametrically controlled over the tip scanning and tunneling current, which opens the possibility for automation.

#### 3.1.2. Research Status

Ternes et al. [27] succeeded in manipulating a single cobalt atom of 160 pm on a Pt and Cu surface via 210 pN. Therefore, many efforts have been made to move a single atom onto the desired workpiece surface with atomic-scale resolution, such as quantum corrals constructed by 48 iron atoms on a Cu surface [28], which is illustrated in Figure 1a. Repp et al. [29] realized the manipulation of turning a neutral charge-state gold atom into a negative charge-state gold ion using a tungsten tip between 5 and 60 K. This breakthrough made it possible for memory devices to store every single bit of information on a per-atom basis. Additionally, the close-to-atomic scale SPL technique has also been extended to molecular manipulation. Quek et al. [30] employed STM with a gold tip to touch a gold surface adsorbing 4,4′-bipyridine and found that a single-molecule junction could be created, which could be turned on and turned off with a stretching or compressing operation by the gold tip. This discovery gave a fundamental understanding of the molecular resistance that can be applied in future molecular electronic devices as a molecular switcher, which is shown in Figure 1b.

Kawai et al. [31] employed AFM to perform the lateral and vertical manipulation of a defect Br^−^ ion on an NaCl (100) surface. With regards to lateral manipulation, as shown in Figure 2a,b, the Br^−^ ion was manipulated three times (marked with numbers 1, 2 and 3): exchanging between Br^−^ and Cl^−^ on the NaCl (100) surface along with [100], diagonal exchanging between Br^−^ and Cl^−^ on the NaCl (110) surface along with [110], and moving Br^−^ to an unimaged area along with [110]. This Br^−^ lateral manipulation is a more complicated atom removal process compared to the general adatom lateral movement, which has been reviewed elsewhere [32]. The Br^−^ vertical manipulation was transferred via picking up and subsequent implanting processes, as shown in Figure 2c. The Br^−^ was picked up, as indicated by an abrupt signal change in the oscillation frequency when the tip was implemented with an approach–retraction operation. Then the tip carrying the Br^−^ was moved to the desired position and the Br^−^ was implanted onto NaCl surface, confirmed by a sudden decrease in the oscillation frequency signal. Finally, a “Swiss cross” atomic-scale structure consisting of 20 Br^−^ ions was created by a repeated operation, which is demonstrated in Figure 2d.

AFM is a powerful tool not only for atomic manipulation at room temperature but also for monitoring and tracking individual electrons in a defined artificial atomic structure. For example, Mohammad et al. [33] first erased a hydrogen atom using AFM on a hydrogen-terminated silicon surface to create six silicon dangling bonds, as shown in Figure 3. Then they observed how the electron jumped between the defined artificial atom structure. This technique was the first step in developing atomic circuits in the future. 

Since the invention of HDL, researchers have engaged in developing this atomically precise manufacturing (APM) approach. Promising results have been reported [34], and the schematics and HDL patterns are shown in Figure 4a,b. To date, HDL has become a variable (not really Gaussian)-spot, variable-energy, vector-scan, e-beam atomic-scale lithography [35]. To enhance the throughput of HDL, Moheimani [36] introduced a new negative sample bias HDL, which simultaneously accomplished lithography and imaging processes. Without switching operations, precision and operational simplicity were enhanced. An automation scheme, voltage-modulated feedback-controlled lithography (VMFCL), was also introduced. Based on VMFCL, experiment results proved that the arrays of depassivated hydrogen atoms can be fabricated. Besides, Rashidi [37] introduced the deep learning-guided HDL method, aiming to detect defects before atomic-scale manufacturing. A convolutional neural network was trained to distinguish common defects on a silicon surface using STM. With the augmentation in autonomous tip shaping and patterning modules, minimal user intervention can be attained for atomic-scale lithography. One important application of HDL in quantum devices and electronics is through atomical precision placement of dopants on the depassivated semiconductor surface [38]. When a depassivated site on a silicon surface is exposed to dopants, atoms like phosphine becomes attracted to the site because of the unsatisfied bond. In this way, HDL could be used to image and place phosphorus dopant atoms in atomically precise locations on a silicon surface. Afterward, the silicon epitaxy technique will encapsulate them with layers of atoms. Except for silicon-based substrate, Scappucci et al. [39,40] used the same method on germanium-based materials, and reported patterns with feature sizes from 200 nm to 1.8 nm. Apart from H-saturated surfaces, alternative resists like halogen atoms, native oxides [41], and molecules [42] have been used to manufacture atomic devices [43]. Halogens have attracted the most attention, and promising results have been demonstrated in recent studies [44,45]. Similar applications with HDL could be seen in the manufacturing of next-generation nano devices such as single-electron transistors [46], quantum computer qubits [47,48,49], editable atomic-scale memories [50], and two-dimensional quantum metamaterials [51]. Considering the throughput of HDL as a largest barrier to commercial use, Randall et al. [52] explored the feasibility of enhancing the throughput via a straightforward, highly parallel exposure system with sub-1 nm resolution. They developed a MEMS-based control scanner with three degrees of freedom (3 DoF) of movement in which they were able to place 7 × 106 tips on a 300 mm wafer. However, this technique still faces engineering challenges during patterning, along with reliability and automation. 

#### 3.1.3. Merits and Drawbacks

In summary, the close-to-atomic scale SPL technique shows excellent capability in changing mechanical, electronic, and chemical properties of a specimen surface to achieve local repair of the workpiece without destruction via a simple single-atom manipulation and molecular transformation process, which highlights the future atomic-scale machining research direction. However, the process requires a top-end conductive tip and workpiece, a vacuum operation environment, and fixed temperature. Besides, close-to-atomic scale SPL shows a lack of efficiency. Single-tip HDL could achieve sub-nm atom removal at around 104 atoms/s [52], which is even less efficient than conventional electron beam lithography. On the other hand, HDL needs further improvement in operational reliability to fulfill the atomic manufacturing requirement of electronic and quantum devices.

### 3.2. O-SPL Nanofabrication Approach

#### 3.2.1. Fabrication Mechanism

The oxidation-scanning probe lithography (O-SPL) approach was first conducted in 1989 by the National Institute of Standards and Technology [53]. The processing principle of O-SPL is shown in Figure 5. To generate a nanoscale oxidation structure, an anodizing reaction between the probe and the substrate surface is adopted. In the oxidation process, the AFM probe and the sample surface are maintained as the cathode (negative) and the anode (positive) of the electrochemical anodic reaction, respectively. Whenever the OH^−^ ions (hydroxyl ions) are needed, the molecule of water gets attached to the sample surface to serve as the electrolyte in the oxidation reaction [54]. The major contribution of O-SPL is achieving a liquid meniscus bridge between the probe and the targeted surface. The most interesting aspect of this process is the generation of multiple bridges. In terms of the simulation method, MD simulation is a reliable and effective tool to guide the transformation process of this bridge on the silicon surface, and the results indicate that the formation is time-efficient [55]. The bridge size can be enlarged by enhancing the voltage strength from 20 to 30 volts, and the range of the pulse duration can be from 10 μs to 10 s [56]. According to the discussion, the size of this bridge can ensure the resolution of the targeted structures. Moreover, O-SPL can also be carried out under contact mode or non-contact mode. 

#### 3.2.2. Research Status

Currently, O-SPL is recognized as a key technology to facilitate the development of the nanoelectronics industry. It has been commonly utilized in resist masks [58], graphene, semiconductors, polymers, metals, and thin conductive films for nanoscale patterning [59], and has been successful in fabricating nanoscale functional devices such as field-effect transistors, single-electron transistors, and single-electronic memory devices [60]. For example, graphene nanoribbons with neat edges can directly be prepared by O-SPL, as illustrated in Figure 6. The reason for the neat edges of graphene nanoribbons can be attributed to the gaseous oxide diffusion in oxidized graphene in the air [61].

Martinez et al. [62] utilized O-SPL to accomplish a single crystal silicon field-effect transistor with a nanoline structure. Figure 7a shows the partial cross-section of the silicon nanoline transistor and electrodes on both sides of the transistors; the width of the nanoline is nearly 9.5 nm. Figure 7b shows high-accuracy sub-20 nm straight and round nanoline structures achieved by O-SPL. The nanoline transistor succeeded in immunological examination application and an on/off current ratio of up to 10^5^. 

Additionally, a qualitative investigation was developed for dichalcogenides for transition metal application. Espinosa et al. [63] carried out the fabrication on MoS_2_ substrate and obtained a 200 nm nanochannel successfully with barriers of around 30 nm, as demonstrated in Figure 8. The electrode and base were gold and silicon dioxide, respectively. The result shows that the electrons could flow in the fixed nanochannel. More importantly, the overall conductivity of the MoS_2_ was not weakened. This means that the size of the conductive channel was downscaled from micrometer scale to nanometer scale. Additionally, transistors depending on MoS_2_ can have wide application in nonvolatile memory cells [64] and medical biosensors for cancer-sensitive identification [65]. Another example came from the work of Dago et al. [66], who obtained 1 nm height and sub-20 nm width and 40 periodicity nanodots structures on the multiple layers of WSe_2_, as shown in Figure 9. These results indicate that the O-SPL is envisaged as a straightforward approach for 2D transition metal fabrication. 

A key study in terms of the slight possibility of 3D fabrication was reported by Lorenzoni et al. [67]. Figure 10 shows the high aspect ratio array nanodots obtained with the highest nanodot above 100 nm via changing voltages up to 10 V and selecting a pulse time between the conductive tip and 6H-SiC substrate. The figure illustrates the new fabrication possibility for 3D nanostructures. 

Interestingly, the O-SPL can be used to coat the silicon surface by embedding nanoparticles [68]. Figure 11a–d shows the entire embedded nanoparticles process. The CoFe_2_O_4_ nanoparticles were successfully embedded in the SiO_2_ layer by using O-SPL and eventually dot and line structures, as shown in Figure 11e,f.

#### 3.2.3. Merits and Drawbacks

The oxidation process is simple and easy to use, and the fabricated structures were extremely stable and robust. The oxide material has the characteristics of insulation and corrosion resistance and was seen as compatible with the existing nanoelectronics machining process. This method can create a mask with high hardness while operating in a low pressure range, which can effectively avoid the proximity effect of electron beam machining. Additionally, the widespread academic use of O-SPL is from metals to semiconductors and more recently to graphene and polymers. Moreover, the technical operation is under room temperature and atmospheric pressure, making O-SPL appealing for academic research. In spite of the material patterning diversity, O-SPL has high requirements for the oxidation of the sample material, which limits the application scope of this technology to some extent. Additionally, it is difficult to obtain high-accuracy nanostructures in a large area due to the drift, hysteresis lag, and nonlinearity problems of the piezoelectric actuator of the SPM itself. It has been very challenging to fabricate complex 3D nanostructures with controllable depth using this method.

### 3.3. T-SPL and Tc-SPL Nanofabrication Approach

#### 3.3.1. Fabrication Mechanism

Although thermal scanning probe lithography fabrication (t-SPL) and thermochemical scanning probe lithography fabrication (tc-SPL) are conceptually simple, they are flexible and convenient scanning-probe lithography methods. To form topographic structures, the thermal process can achieve desired material removal, and the processing is referred to as t-SPL [69]. Accordingly, this method usually modifies the polymer mechanically. For instance, some researchers have utilized a substrate called transparent polymethyl methacrylate (PMMA) to conduct nanoindentation aiming at high-density data storage device fabrication [70]. Most importantly, a conductive substrate surface is not the necessary requirement for t-SPL [71]. The achieved structures of tc-SPL are essentially completely thermochemical and consist of a certain type of material that has the various conformation and chemical compositions compared to the original structure [72]. In this method, the cantilever of the resistance-heated SPM induces a chemical reaction and modifies material surface functionality. It has been noticed that, in the t-SPL and tc-SPL methods, the heat can be considered a pivotal function part. Especially for the tc-SPL method, the chemical reaction rate ramps up along with the accumulation of heat. In the early stage, the laser can be used to heat the cantilever. However, it was still a challenge to combine the heating laser source with the designed SPM system. Currently, cantilevers integrated with resistive heaters are emerging as a potential technology with the advantage of easier integration. As shown in Figure 12, it can reach a temperature over 1000 °C depending on the type of dopant, and the thermal time constant can be fast—up to 10 µs [73]. 

#### 3.3.2. Research Status

The t-SPL and tc-SPL targets are used to machine a wide range of materials, such as molecular glass resist [75], biomaterials, organisms [76], 2D materials [77,78], metals, carbon nano-tubes [79], nanoparticles, and polymers, including supramolecular polymer [80], polycarbonate [69,81], polystyrene [82], block copolymers [83], and polyethylene [84]. Representative work of the t-SPL was carried out in the early 1990s, and it was first extended to the area for data storage purposes. In these works, multiple heated tips were utilized to pattern the pit structures into a polymeric compound substrate. The data bits written on a polycarbonate substrate spaced less than 200 nm apart with a patterning time of 5 ms per bit have been achieved [69]. Moreover, t-SPL has been widely utilized to create a nanostructure on the Si substrate with high resolution. It has also verified a lower line-edge roughness, as shown in Figure 13. Furthermore, Cho et al. [85] revealed the optimal mechanical force of 25 ± 6 nN and the best heated tip temperature of 550−700 °C when fabricating nanopatterns on the Si substrate.

To date, the t-SPL method has been used for the fabrication of quantum nanoscale electronic components. For example, Rawlings et al. [87] employed a hybrid method including t-SPL and laser machining to obtain a single-electron transistor with a 50 nm insulated gate. Figure 14a,b shows the process of the polypthalaldehyde (PPA) patterned by t-SPL with 15 nN scratching force under 950 °C and the patterned result, respectively. Figure 14c demonstrates the nanostructure finally achieved after t-SPL and laser machining. 

In fact, the t-SPL is a clock concept paired with heating the tip of the SPM. Shaw et al. [88] conducted experiments by heating the tip and substrate up to 500 °C simultaneously to fabricate PMMA and Pentacene at 120 °C, 140 °C, and 160 °C. The relationship between the substrate temperature and maximum scanning speed is shown in Figure 15. These selective high temperatures did not make the substrate material undergo phase transformation. It was found that the fabrication speed was up to 19 times faster than the conventional fabrication by only heating the tip of the SPM. Recently, another important application of the t-SPL is in the anti-counterfeit symbol field. Samuel et al. [80] explored the nanostructure of fluorescence features on the supramolecular polymer with thermochromism using the thermal tip characteristic of t-SPL, which could be applied in counterfeit security. The color of the supramolecular polymer changed from red to green when the heated tip contacted the local area. 

As for tc-SPL, a conjugated polymer of *p*-phenylene vinylene (PPV) is quite an active substrate due to its electroluminescent feature in this approach, which is widely applied in the field of nanophotonics and light-emitting diodes (LEDs) [89]. For example, Wang et al. [70] used tc-SPL to obtain a 70 nm-width nanoline structure on a PPV substrate of 100 nm thickness with a 240 °C hot tip. The scanning speed and vertical loads were 20 μm/s and 30 nNs, respectively, as shown in Figure 16. Oliver et al. [90] succeeded at running the finite element model simulation process of the thermal tip approaching the PPV surface and indicated that the nanoline achieved more accuracy when the PPV layer became thinner and the radius of the tip was smaller. 

#### 3.3.3. Merits and Drawbacks

It has commonly been accepted that the t-SPL and tc-SPL are simple, direct, and extremely effective fabrication technologies for simple small patterns with a fast scanning speed [91]. These methods not only achieve sub-20 nm fabrication accuracy [92] but also reduce the probe wear effectively. The focal point of the tc-SPL method is to work in wet environments. In other words, water film structures of several nanometers could be realized in humid environments. Unfortunately, these methods are only suitable for patterning highly thermosensitive materials in high resolution. The existing challenge that still needs to be solved for these methods is how to evaluate the heat flow through the tip and how to predict the interface temperature. Simultaneously, the thermal effects limit the tip’s apex size. 

### 3.4. D-SPL Nanofabrication Approach

#### 3.4.1. Fabrication Mechanism 

Dip-pen scanning probe lithography fabrication (D-SPL) is a novel method that employs a kind of “ink”-coated AFM probe to machine the substrate through electrostatic interactions [93] or electrochemical reaction [94]. Dip-pen SPL is a direct-writing nanofabrication method that can pattern soft and hard materials from a scanning probe onto a sample surface with accurate position and sub-100 nm resolution [95], just as ink moves from a visible ink pen to paper. Figure 17a demonstrates the mechanism of the approach. Importantly, D-SPL is compatible with various inks, such as organic molecules [96], polymers [97], proteins [98,99], inorganic nanoparticles [100], DNAs [101], and metal ions [102]. Significantly, the ink molecules can be transported to the tip through a microfluidic channel so that the ink needed in the process can be supplied continuously, as illustrated in Figure 17b. Additionally, a heated AFM probe can be used in D-SPL. It is known as thermal dip-pen scanning probe lithography [103]. As a variation of D-SPL, thermal dip-pen scanning probe lithography can control and transport solid or insoluble inks at room temperature. 

#### 3.4.2. Research Status

To date, D-SPL has undoubtedly been developed to be multiplex [106]. Firstly, based on the above-mentioned prototype method, scholars directly deposit target material onto the substrate surface by solvent instead of transferring by water meniscus, where the target material is usually small molecular material and can only be soluble in the water meniscus. This improved approach expanded the range of inks enormously. For example, Hung et al. [107] reported that an Ag nanoline structural height ranging from 120 nm to 260 nm with an averaging resistor of around 30 μΩ·cm was accomplished by depositing the Ag material nanoparticle solvent onto the SiO_2_ substrate. The process mechanism schematic and the image of the Ag nanoline structures are shown in Figure 18a–c, respectively. This method offers a specific metal nanoparticle deposition technique that can be widely applied in the field of printable electronic and electronic invalidation analysis.

Secondly, a matrix can be exploited to assist in depositing target material onto the substrate surface through water meniscus or solvent, in which the matrix is a carrier. For example, Chen et al. [108] contributed by developing a precursor for the formation of multimetallic nanoparticles, downscaling to the nanometer level. In this work, the PEO-*b*-P2VP was made to be the matrix carrying the five kinds of metal nanoparticle, which were Au, Ag, Cu, Co, and Ni, or a mixture of them, and then a sub-10 nm polymetallic alloy hemisphere nanostructure was created. 

Additionally, Nelson et al. [109] expanded the D-SPL to the thermal D-SPL by heating the tip similar to t-SPL and tc-SPL and obtained a sub-80 nm-width line structure on the glass substrate mixed with borosilicate. The local deposition of the line structure was an indium metal coating on the tip, as demonstrated in Figure 19a. This technique provided a novel approach for the circuit repair, such as a gap of around 500 nm being repaired by thermal D-SPL between both sides of the gold electrode, as shown in Figure 19b. The aim of the thermal application is to control the ink regardless of the melting of the indium metal. Therefore, the deposition could be controlled easily. Similarly, 6 nm-high nanoparticles consisting of Fe_3_O_4_ were achieved [110] using thermal D-SPL. 

#### 3.4.3. Merits and Drawbacks

D-SPL not only deposits a variety of nanoparticles and nanocomposites but also is a real maskless nanofabrication approach because the processed substrates do not require retreatment, such as further tailoring or a solution technology process. Moreover, in this method, the achieved line width is independent of the speed of the tip and the contact force. Therefore, when a parallel and multiple tip fabrication system is carried out [111,112], only one tip is controlled by the feedback control system and other tips can perform the same actions. Correspondingly, a large number of nanoline structures and even a wide range of 2D nanostructures are able to be created and the fabrication efficiency can be improved rapidly. However, the local deposition has the drawback of being inhomogeneous and inconsecutive, especially for nanoline structures. In addition, while filling the nanoparticles in the polymer, there is a composite thermodynamically stable problem caused by the radius of the nanoparticle when its radius is bigger than the radius of the polymer [113]. That is to say, the approach has limitations based on the size of the inks. 

### 3.5. B-SPL Nanofabrication Approach

#### 3.5.1. Fabrication Mechanism

Bias-induced scanning probe lithography (B-SPL) is derived from applying a bias voltage between the SPM tip’s apex and substrate surface. The implemented voltage, which is usually from 0 V to 20 V, plays a pivotal role in the fabrication process. When applying about 0 V to 20 V, the real electric field is from 10^8^ Vm^−1^ to 10^10^ Vm^−1^ between the tip and substrate surface. The B-SPL can generate different chemical and physical fabrication results on different substrate materials. For example, Figure 20 [114] shows the mechanism of the generation of nanoscale structures on the extra-thin polymer film substrate surface by employing a high-electricity conductive tungsten carbide tip. Accordingly, the focused electron current brought by the high electric field will achieve a gradient distribution electric field configuration and make the polymer film surface turbulent and polarized. Furthermore, the current density can vary almost linearly with the applied bias voltages. It was discovered [115] that when a tip with a radius of less than 30 nm approached the polymer surface, the actual distance was less than 5 nm between the tip and polymer surface. The surface of the polymer would bulge so that the height became10 nm to 50 nm [116]. Subsequently, nanoline structures and diverse 2D nanostructures could be accomplished with the moving tip. Therefore, it is a physical fabrication process for processing polymer materials. In addition, B-SPL is also utilized to disintegrate gas or liquid molecules to form sediments to modify a substrate surface, which is a local nanochemistry reaction deposit process motivated by focused electron current. 

#### 3.5.2. Research Status

To date, the B-SPL technique has been widely implemented to fabricate a variety of polymers with accuracy ranging from 10 nm to 100 nm nanostructures for versatile applications such as molecular electronics [117], nanoscale sensors [118], and data storage [119]. Apart from this, the B-SPL method has been expanded to other samples, such as NaCl thin film covered on an Au substrate surface [120,121], achieving GaAs/AlGaAs nanoline heterostructures on an Si substrate surface [122], and various solvent liquids, including alcohol, dioxane, and octane [123,124,125]. Additionally, an electronic current created by bias voltage is used to limit various chemical reactions and to decompose the deposition of gases or liquid molecules or the growth of materials on the surface. For example, Garcia et al. [126] obtained sub-25 nm accuracy for carbon nanodot structures on an Si substrate surface by employing the B-SPL method to transfer the CO_2_ gas under voltage ranging from 10 V to 40 V, as shown in Figure 21a. Figure 21b,c show the nanodot structure fabricated under different processing times with 21 V and different voltages under 0.1 ms, respectively. Moreover, the nanodot structure could be extended to the scale of the square centimeter area. Another key example of this method is the periodic lines with a pitch of 1 μm, 80 nm width, and 0.32 nm height obtained by using a high-conductivity tungsten carbide K-TEK tip across a thin polymer film and applying 18 volts [114].

#### 3.5.3. Merits and Drawbacks

The B-SPL method can employ the electric field to realize the nanoscale fabrication of polymer film surface topography. For processing polymers, it is a physical fabrication process, and no chemical reaction exists. The polymer will not be degraded nor will it have any abrasion. The polymer is only to be transported under the effect of a high electric field and then achieves the nanostructure without the external thermal source. The B-SPL method is quite suitable for creating the nanostructures in a polymer film surface, yet the achieved product is not easily stable or homogeneous. Due to the decomposition of gas and solution molecules, this process has expanded itself to the nanochemistry domain. The deposition induced by the electrochemical reactions can be created on various substrate surfaces and is more stable and robust. Despite this advantage, the deposition process is time-consuming and the throughput is low. Furthermore, the mechanism of controlled nanochemistry reactions and bias voltage between the tip’s apex and substrate surface has not been fully understood yet.

### 3.6. M-SPL Nanofabrication Approach

#### 3.6.1. Fabrication Mechanism

Mechanical scanning probe lithography (M-SPL) prompts the selective removal of materials from the substrate surface by several nN mechanical forces applied at the probe using ploughing, milling, and cutting via atomic-force microscope (AFM) [127,128,129]. To sum up, this technique can be categorized into two work types depending on the AFM scanning mode. When AFM works in the contact mode, the interaction force between the tip and substrate surface with a magnitude can range from 10^−8^ to 10^−11^ N, which gets enlarged by adapting a larger cantilever deflection. The tip acts as a cutting tool. With enough normal force, it is capable of inducing plastic deformation in the substrate surface and then removing material through further shearing. During this nanofabrication process, the tip is kept in translational motion by presetting the normal force and program. This process is called static ploughing lithography, as shown on the left side of Figure 22. When AFM works in the tapping mode, as illustrated on the right side of Figure 22, a bigger amplitude applied on the cantilever makes the cantilever achieve its resonant frequency. Subsequently, the substrate surface can be modified by a continuous hammering or indenting process compared to the shearing nanofabrication in the static ploughing lithography. This process is, therefore, called dynamic ploughing lithography. Both AFM working modes enable the M-SPL approach to operate easily by directly writing on the task workpiece surface. 

#### 3.6.2. Research Status

To date, the M-SPL approach has been applied in patterning metals, semiconductors, and polymers. Researchers have employed this method to obtain 2D and even 3D nanostructures. At present, static ploughing lithography working under AFM scanning mode has been broadly employed in early nanoscratching experimental investigations, in which a diamond tip with a highly elastic spring constant up to 100–300 N/m and a range of 30–50 nm [131] probe radii is usually used to fabricate nanoscale structures. For example, many scholars initially employed a piezoelectric actuator and the so-called closed-loop system of the AFM to fabricate arbitrary nanostructures by means of controlling different feedback gains and scanning speeds. Wang et al. [132] employed M-SPL to investigate the patterning of nanostructures with a desirable machined depth on GaAs material. Hyon et al. [133] achieved a sub-20 nm-width nanoline with nearly 1 nm depth on the GaAs surface. Wendel et al. [134] succeeded in obtaining a 16 nanoscale hole array with a 55 nm periodicity GaAs/AlGaAs heterostructure substrate under ambient conditions, as shown in Figure 23a. Furthermore, they also accomplished a smaller array of 35 nm nanoscale holes. The diameter of these holes could be several nanometers. Afterward, Schumacher et al. [135] continued to use this approach to machine GaAs/AlGaAs heterostructures with 50~100 μN contact force under 100 μm/s scanning speed to fabricate a channel barrier and insulated gate, which is illustrated in Figure 23b. These works opened up the application market for single-electron transistor fabrication with single-gate and quantum electronic nanocomponent integration production. The M-SPL approach not only patterns on the target substrate surface directly but can also be combined with other nanolithography techniques, such as the lift-off process, wet chemical etching, and dry etching, achieving the purpose of machining nanostructures on various material surfaces. Figure 23c shows a single-electron transistor fabrication process using the mixture of the M-SPL approach, dry etching, and the lift-off process [136]. 

It can only pattern nanostructures with widths ranging from 10 nm to 100 nm and depths ranging from 1 nm to 4 nm. It is harder to realize complicated 2D and 3D nanostructures. Accordingly, Lee et al. [137] put forward a system similar to an AFM using the diamond tip to scratch the material surface, achieving regular and complex structures. However, these structures have low micrometer accuracy owing to the bigger mN force exerted by the system and the bigger tip radius of up to several micrometers [138]. In view of this, one longitudinal study by Yan et al. [139] came up with a CNC nanoscale 3D worktable that was based on a commercial AFM and a high-precision stage. The high-precision stage was used to control the accurate movement by another external computer. During the fabrication process, the tip was fixed onto the substrate surface with a preset force and then the high-precision stage started to move according to the preset requirement. As a consequence, complicated 2D and 3D nanostructures were obtained [139]. 

A seminal study in this research area is the work of 3D patterns. M-SPL is similar to the traditional cutting process, which can realize the processing of 3D nanostructures. The specific characteristic of M-SPL is its capability to precisely control the machining parameters at the nanometer scale and microsecond timescale during the patterning process. A 3D polymer bundle structure similar to the traditional sinusoidal structure (see Figure 24) was obtained using a single crystal silicon cantilever with an elastic spring constant of up to 200 N/m on the polycarbonate (PC) surface through a single scan. Additionally, a greyscale 3D human face within a frame size of 20 μm × 20 μm was written on a polished Al disk sample with a 9.8 nm Ra in less than 10 min [140]. Moreover, Mao et al. [141] accumulated the sample material on one side by controlling the trace of the probe to form a 3D structure. Based on this method, a 3D micro-Taiwan island pattern was successfully fabricated (see Figure 25). 

In contrast to static ploughing lithography, dynamic ploughing lithography has drawn more researchers’ interest in recent years for the creation of 3D features and repair of advanced lithographic masks, as it eliminates the lateral force effect, which can cause subsurface damage, and has the advantage of being non-ridged for nanogroove fabrication. So far, this technique has been focusing on polymer, graphene, and metal patterns by using silicon or a silicon nitride cantilever with an elastic spring constant of 10–100 N/m and a probe radii ranging from 10 to 30 nm. For example, Figure 26 demonstrates that various 3D nanodots can be fabricated on the soft PMMA thin film by employing the dynamic ploughing lithography approach [143,144]. In addition, Borislav et al. [145] used dynamic ploughing lithography to tailor and manipulate the geometry of graphene, avoiding the disadvantages of uncontrollable crumbling, dragging, and ripping from static ploughing lithography. Meanwhile, Yan et al. [2,146] carried out dynamic ploughing lithography experiments on single-crystal Cu and found that the machining direction of the tip significantly influenced the depth and pile-up on the sidewalls of nanogrooves. Furthermore, Xiao et al. [130] compared static ploughing lithography and dynamic ploughing lithography for machining Cu and revealed that less chip formation and smaller feature sizes were observed in the dynamic ploughing process.

#### 3.6.3. Merits and Drawbacks

In brief, M-SPL is a promising method for advanced nanofabrication in an AFM. The method is simple to use and possesses excellent control ability, with the flexibility to operate under the conditions of ultra-vacuum, atmosphere, liquid, low temperature, normal temperature, and high temperature, and does not require complicated machines. In addition, the size of mechanical machined structures can approach nanoscale. Nonetheless, the limitation in the fabrication of consistent structures stems from the stability of the probe caused by its deformation and contamination due to the debris of the removed material. Inevitably, M-SPL, as a mechanical machining approach, results in subsurface damage (SSD) in the cutting zone, degrading the component life. Normally, a post-machining operation is a requirement to remove the SSD, such as chemical etching or chemo-mechanical polishing. In addition, when certain nanostructures need to be produced in mass production or require a smoother surface, such as a refined 3D sphere surface, the M-SPL method must be combined with other nanofabrication methods, like electron beam lithography (EBL), the nanoimprinting lithography technique (NIL), or focused ion beam fabrication (FIB). Moreover, the probe of AFM will wear rapidly when machining some semiconductor materials. Finally, the mechanism of atomic scale material removal requires further studies. 

### 3.7. New SPM Tip-Based Nanofabrication Approaches

The flexibility and versatility of scanning probe microscopy to direct writing and deposition on surfaces have led to the creation of some other methods, such as the SPM tip-based dispensing approach, the SPM tip-based ultrasonic vibration-assisted approach, and the SPM tip-based magnetic approach. Dispensing based on SPM fabrication utilizes a cantilevered nanopipette, which is hollow, to replace the probe of a scanning probe microscope. The jet-flow tube can directly deliver soluble molecules to any surface. For example, protein featuring as small as 200 nm [147] and single living cells under physiological conditions [105] can be dispensed to the sample surface by using the dispensing feature of the SPM tip-based fabrication method. Deng et al. [148,149] employed the ultrasonic vibration technique to complement the SPM tip-based nanofabrication. They kept the high-frequency vibration in the XY work stage and high ultrasonic vibration in the Z tip direction. They obtain 3D concave and stair-stepping nanostructures on the PMMA surface within several minutes. Additionally, closed-loop electric field current-controlled scanning probe lithography has been employed to assist in electron beam lithography to implement mix-and-match lithography with sub-10 nm resolution on molecular glass resist [150]. Recently, a magnetic approach by SPM tip-based nanofabrication has been to use a magnetic force microscope (MFM), which helps microstates access the artificial spin ices (ASI) and relevant non-interacting nanomagnet arrays by writing the topological defects into magnetic nanolines directly [151]. Much more interestingly, a novel tip-based electron beam-induced deposition (TB-EBID) technique has recently been proposed to fabricate single-electron devices with a low-energy beam [152]. 

## 4. Application of SPL Nanofabrication Technique

### 4.1. Nanofluidic Science

Nanofluidic channels play a pivotal role in the field of nanofluidic science, which can offer a physical confinement environment to manipulate and analyze DNA [153] and single molecules [154]. Since the birth of the SPL nanofabrication technique, it has emerged as a rapid and flexible approach to fabricate arbitrary structures of nanofluidic channels in comparison with the previous expensive and complex EBL and FIB methods. For example, Hu et al. [155] utilized the SPL nanofabrication technique to directly fabricate an etch mask by depositing polymer nanowires on the Si surface. The nanostructures on the Si surface via single step etching were employed as a mold for the mass production of polydimethylsiloxane (PDMS) nanofluidic channel. Furthermore, a PDMS nanofluidic channel with both straight and curvilinear structures was fabricated by utilizing the SPL nanofabrication technique, as illustrated in Figure 27. Meanwhile, the SPL nanofabrication technique showed remarkable compatibility with the current nanofabrication approach.

### 4.2. Biomedical Application 

A critical application of the SPL nanofabrication technique is to characterize the mechanical, physical, and chemical properties of cells, proteins, scaffolds [156], and 2D biological tissue/thin film [4]. Researchers employed SPM to trigger nano-indentation to establish the indentation model, the so-called Hertzian contact model, by extrapolating the mechanical compliance between tip and specimen. To this end, the elastic–plastic deformation related to linear elastic deformation of the uploading curve was measured [157]. Additionally, several physical chemistry reactions occurring at the interface between cells and scaffold were studied. Subsequently, the SPL nanofabrication technique was applied to modify the surface of the scaffold to govern cell response. Moreover, the SPL nanofabrication technique was referred to in order to have the capability to deliver nanoparticles and nanofibers using the tip as a drug carrier [158].

### 4.3. Quantum Computing and Data Storage Device 

The technique of close-to-atomic scale SPL has been applied in various research aspects, such as quantum dot machining and single-atom data storage device machining. For example, Stefan et al. [159] employed STM to create quantum dots of single-atom precision fixed by a 2 × 2 In-vacancy reconstructive InAs (111) template surface, which was effective at controlling the position of quantum dots with zero error. The specified location of the quantum dots consisted of a chain of ionized In adatoms moved by using vertical atom manipulation of STM, as shown in Figure 28. 

Cyrus et al. [160] succeeded in storing the data in bits in one magnetic atom. They used the STM atom manipulation technique to place an Fe or Mn atom on the non-magnetic copper-nitride film surface and created a structure of a single magnetic Fe or Mn atom surrounded by non-magnetic atoms, which could align the magnetic moment along one direction and overcome the superparamagnetic limit. Additionally, magnetic anisotropy in just one atom was also observed for the first time. 

### 4.4. Nanoelectronics

Nanoelectronic devices with sizes ranging from 1 nm to 100 nm are essential components for nanoelectronics, such as silicon metal-oxide-semiconductor field-effect transistors (MOSFET), fin field-effect transistors (FinFET), single-electron transistors (SET) [161], and molecular circuits [162]. Lately, several strong applications in the fabrication of SET operating at room temperature (RT) have been presented by using the SPL nanofabrication technique as opposed to the previously presented transistors operating at cryo-temperatures [163]. For example, Durrani et al. [164] employed the SPL nanofabrication technique to fabricate the Si/SiO_2_/Si point-contact tunnel junctions with sub-10 nm size, which could achieve a deeper quantum dot potential well confinement up to 2 eV. The state of the art makes it possible to operate the SET under RT.

## 5. Comparison and Discussion

Table 2 compares the differences in the machining capabilities of various advanced SPL nanofabrication approaches. Close-to-atomic scale SPL is capable of machining atoms with atomic-scale resolution. However, the close-to-atomic scale machining process is slow and requires a complex machining environment. Among these techniques, O-SPL has achieved sub-5 nm fabrication resolution. Due to its high reliability and sub-5 nm fabrication resolution, the fabricated nanostructures can be used as key components of nanoelectronic devices and templates in subsequent etching or deposition work. However, the O-SPL concentrates on the pattern of high oxidation of substrate materials, which confines the application scope of this technology to a certain extent. In terms of tip wear, close-to-atomic scale SPL, O-SPL, D-SPL, and B-SPL have a relatively long tip life. However, D-SPL is a more complicated fabrication technology with very limited control ability [104] to deposit polymer materials or biomolecules onto the substrate. Meanwhile, B-SPL involves the physical process of polymer modification and the nanochemical process for decomposing gas and solution molecules. The throughput of B-SPL is very low because the material transformation and chemical reactions are very time-consuming under high electric fields. In addition, both t-SPL and tc-SPL have less tip wear than M-SPL because the heat effect can soften the substrate material, which makes it easier to cut. In particular, the t-SPL and tc-SPL are more effective at processing thermosensitive materials. As opposed to t-SPL and tc-SPL, M-SPL has the capability to fabricate more wide and diverse materials and breaks through the thermosensitive material limitation in the nanolithography with high control ability. However, the subsurface damage (SSD) in the cutting zone during the M-SPL process can cause fatigue and creep in the machined component. Therefore, a post-machining operation, e.g., chemical etching or chemo-mechanical polishing, is required to eliminate the SSD. Moreover, inevitable tip wear of M-SPL is still a long-standing challenge to overcome. 

## 6. Challenges and Outlook

The SPL nanofabrication technique has already demonstrated remarkable fabrication capabilities for 2D/3D nanostructures, nanocomponents, and even single-atom memory devices. It has been successfully applied in quantum computing, nanoelectronics, and nanofluidics devices. However, there are two major challenges that have limited the commercialization of the SPL nanofabrication technique. 

The first challenge is the low processing efficiency. The SPL nanofabrication technique is based on an SPM platform that is basically designed for measurement purposes in a lab environment. Due to the high precision requirement, the SPL nanofabrication technique is only used for proof-of-principle experiments so far instead of mass industrial production. In other words, it is still a high-value manufacturing method rather than a high-volume manufacturing method. For example, researchers have manufactured quantum wells and single-electron transistors (SET), which only prove that the machining precision of the SPL nanofabrication technique can meet the requirement beyond what is required for making nanoelectronic devices. In order to move towards industrial application, a necessary prerequisite is to enhance the processing efficiency dramatically. To solve this problem, recently, a new SPL nanofabrication strategy has been proposed by using a structured AFM tip [173]. With such a tool prepared by FIB, three-dimensional sin-shaped ripples were achieved with high-precision surface quality [174]. This work proves the scalability of the SPL technique to fabricate nanoscale periodic patterns. Furthermore, a mix-and-match lithography approach by combining the SPL nanofabrication technique and existing nanofabrication techniques, such as wet etching, dry etching, the lift-off process, NIL, FIB, and EBL, will be a better choice to approach mass industrial production [87]. 

Another challenge is the smallest achievable feature. The processing structure is restricted by the size of the region where mechanical interaction or chemical reaction occurs, which is normally directly related to the tip radius. To further reduce the feature size to close-to-atomic scale, the challenge lies not only in the tip size and control and elimination of environmental effects, but also in the fundamental understanding of the manufacturing process, which is based on quantum theory rather than the continuum theory [175]. Therefore, future studies on close-to-atomic scale SPL could focus on the reduction of the interaction region by using sharper tips, reasonable material selection and preparation, environmental control, etc., as well as on the theoretical and simulation study of SPL fabrication to reflect the true determinants of the feature size to allow effective control. In addition, atomic-scale patterns are normally accompanied by weak structural stability due to a low atomic diffusion barrier caused by surface chemical reactivity and structural properties. To allow a stable and long-lasting function, the desired pattern needs special consideration of both materials and atomic structures. On the premise of ensuring processing efficiency, further reduction of the lateral dimension of the nanostructure is a thought-provoking question to realize industrial-scale production. 

## 7. Concluding Remarks 

This paper discussed and summarized the state of the art and future perspectives of scanning probe lithography techniques. The fabrication mechanism, research status, and merits and drawbacks of different SPL approaches were reviewed systematically in this paper. The conclusions drawn are follows:The SPL nanofabrication technique is a unique technique offering low-cost, high-value manufacturing while achieving atomic-scale precision. It offers additional advantages such as not requiring a mask and allows direct writing on the substrate by means of various chemical, physical, diffusive, and deposition mechanisms.The SPL nanofabrication technique has largely been used at the laboratory level to fabricate nanoscale components at a scale envisioned by Nobel Laurette Richard Feynman, and was seen as a long-standing challenge back then even to achieve that level of precision at the nanoscale. With its current success, more efforts are required to enable commercialization of the SPL nanofabrication technique.A mix-and-match lithography approach by combining the SPL nanofabrication technique and the etching technique can pave the way to a cost-effective manufacturing method contrary to the currently used mass nanofabrication production techniques.The SPL nanofabrication technique is a critical nanofabrication method with great potential to evolve into a disruptive atomic-scale fabrication technology to meet the future demand for atomic manipulation of surfaces.

## Figures and Tables

**Figure 1 micromachines-13-00228-f001:**
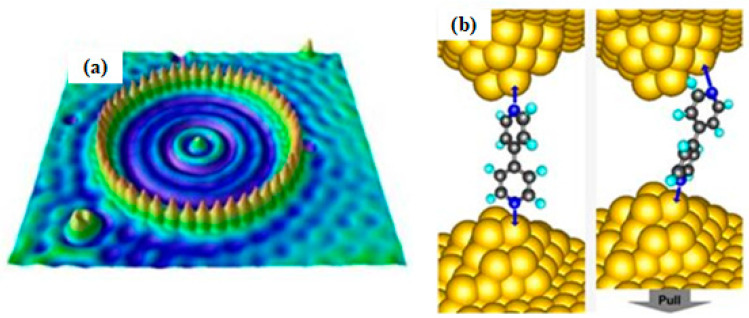
(**a**) STM image of nanometer-scale quantum corrals structure. Reprinted with permission from Ref. [28]. Copyright 2022 American Physical Society. (**b**) A single-molecule junction switcher. Reprinted with permission from Ref. [30]. Copyright 2022 Springer Nature.

**Figure 2 micromachines-13-00228-f002:**
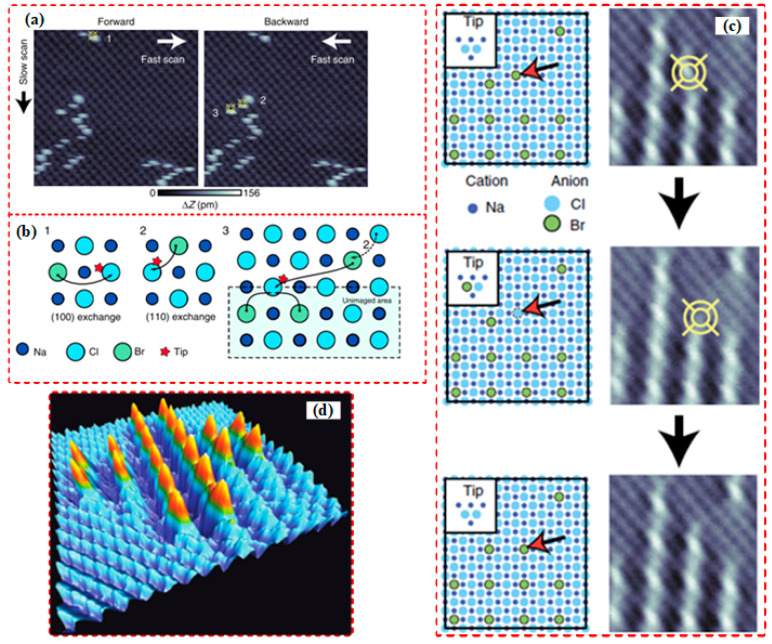
(**a**,**b**) The real forward and backward scanning Br^−^ manipulation sequence event and its theoretical model. The Br^−^ is surrounded by bright topographic features. (**c**) The vertical atom manipulation image process includes before manipulation (top), after picking up Br^−^ (middle), and after implanting Br^−^ (bottom). (**d**) A “Swiss cross” atomic-scale structure image consisting of 20 Br^−^ ions. Reprinted with permission from Ref. [31]. Copyright 2022 Springer Nature.

**Figure 3 micromachines-13-00228-f003:**
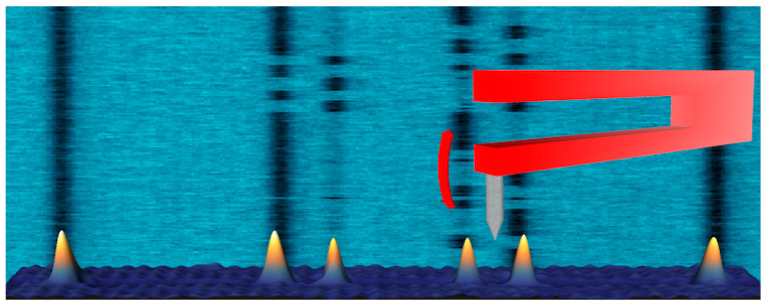
The artificial atomic structure with six silicon dangling bonds. Reprinted with permission from Ref. [33]. Copyright 2022 American Physical Society.

**Figure 4 micromachines-13-00228-f004:**
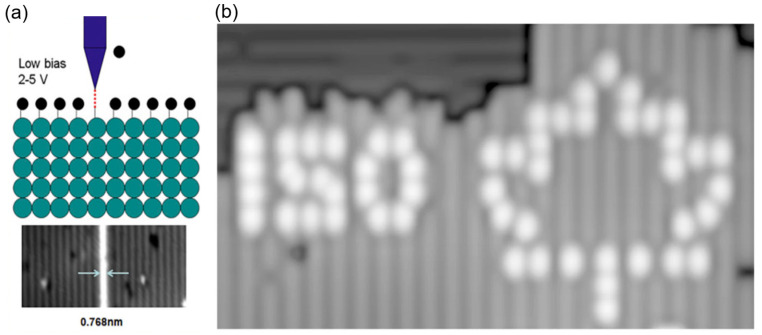
(**a**) Schematics of atomically precise HDL. Reprinted with permission from Ref. [52]. Copyright 2022 American Vacuum Society. (**b**) Fabrication of the characters “150” and a maple leaf using HDL. Reprinted with permission from Ref. [50] Copyright 2022 Springer Nature.

**Figure 5 micromachines-13-00228-f005:**
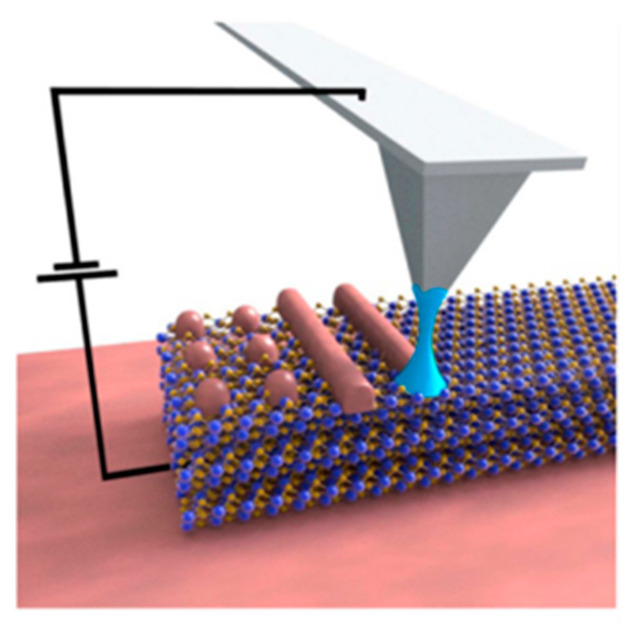
Scheme of nanopatterning using O-SPL. Reprinted with permission from Ref. [57]. Copyright 2022 IOP Publishing.

**Figure 6 micromachines-13-00228-f006:**
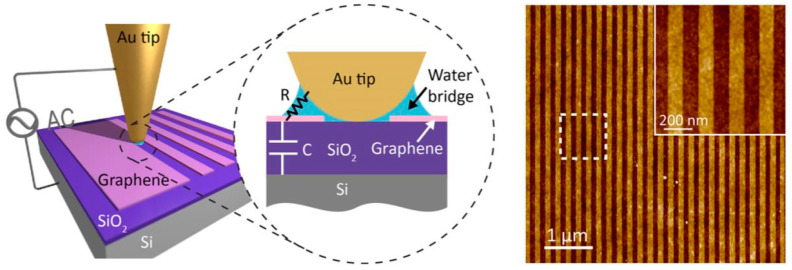
Fabrication of graphene nanoribbons using O-SPL. Reprinted with permission from Ref. [61]. Copyright 2022 American Chemical Society.

**Figure 7 micromachines-13-00228-f007:**
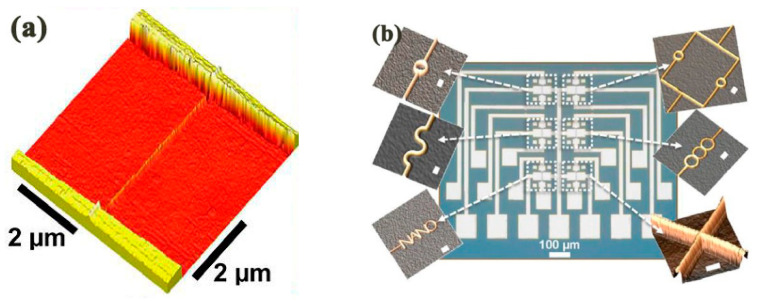
(**a**) The topography of single crystal silicon field-effect transistor using a nanoline nanostructure. (**b**) An image of various sub-20 nm straight and round nanoline structures. Reprinted with permission from Ref. [62]. Copyright 2022 IOP Publishing.

**Figure 8 micromachines-13-00228-f008:**
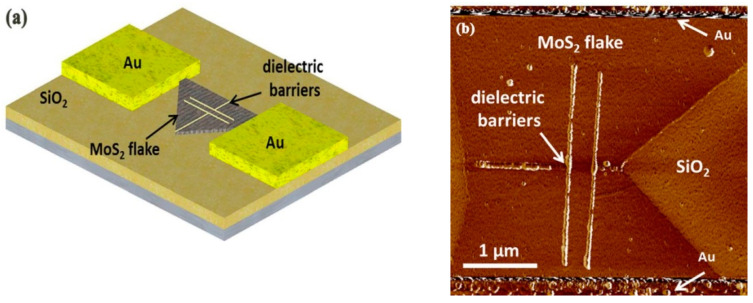
(**a**) The theory model of transistors depending on MoS_2_. (**b**) The achieved nanochannel pattern on the MoS_2_ flake between electrodes. Reprinted with permission from Ref. [63]. Copyright 2022 AIP Publishing.

**Figure 9 micromachines-13-00228-f009:**
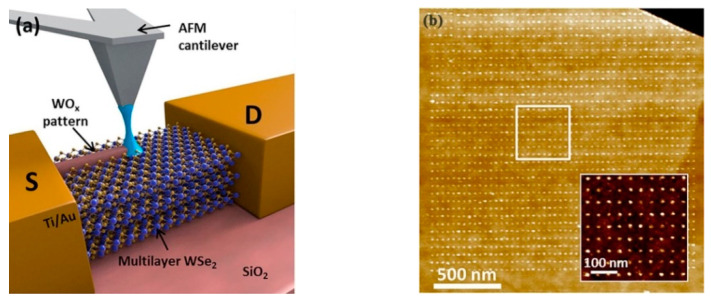
(**a**) The mechanism scheme of the multiple layers of WSe_2_. (**b**) The achieved nanodot oxide structure. Reprinted with permission from Ref. [66]. Copyright 2022 AIP Publishing.

**Figure 10 micromachines-13-00228-f010:**
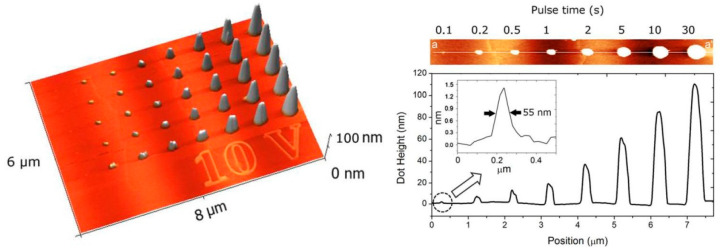
The image of nanodot geometry and height profile. Reprinted with permission from Ref. [67]. Copyright 2022 AIP Publishing.

**Figure 11 micromachines-13-00228-f011:**
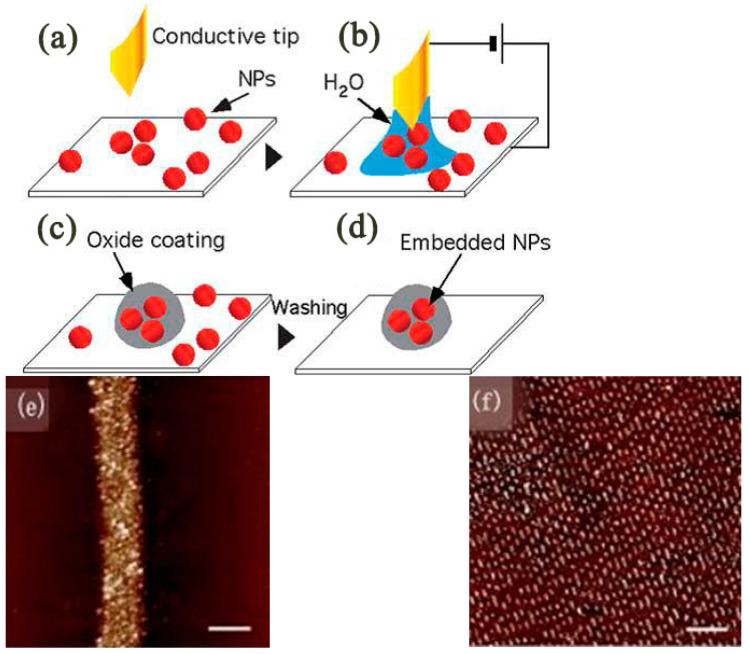
(**a**–**d**) Scheme of nanoembedding. (**e**) Single stripe of embedded CoFe_2_O_4_ nanoparticles. (**f**) The 11 cm^2^ area patterning of a structure containing dots and lines. Reprinted with permission from Ref. [68]. Copyright 2022 RSC Pub.

**Figure 12 micromachines-13-00228-f012:**
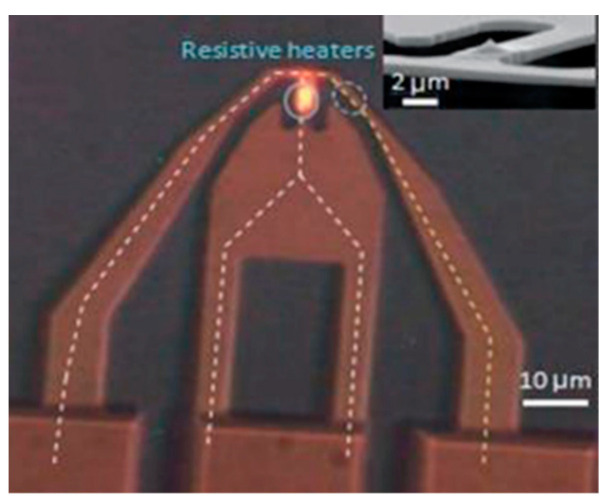
A resistive heating AFM cantilever with two highly doped silicon legs. Reprinted with permission from Ref. [74]. Copyright 2022 Springer Nature.

**Figure 13 micromachines-13-00228-f013:**
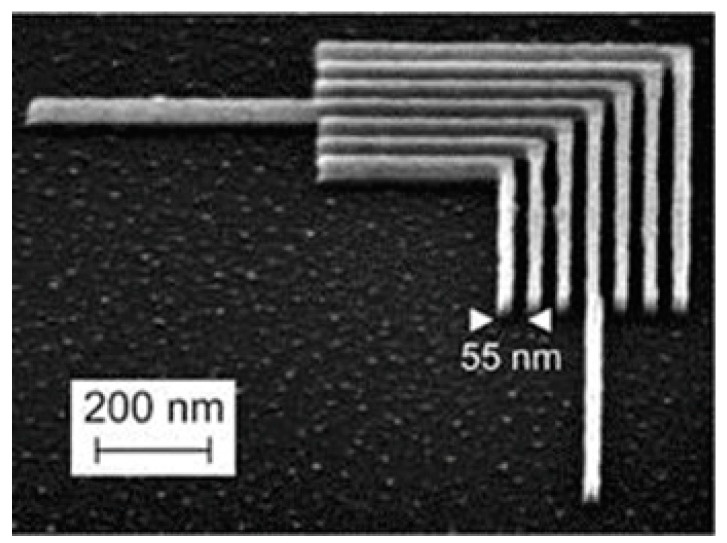
L-line structures at 27-nm half-pitch. Reprinted with permission from Ref. [86]. Copyright 2022 American Chemical Society.

**Figure 14 micromachines-13-00228-f014:**
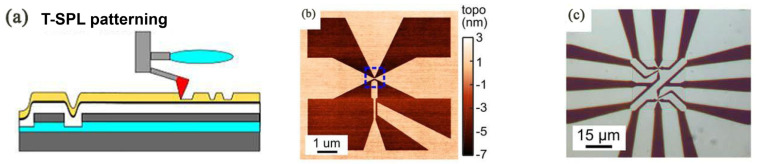
(**a**) The process of the polypthalaldehyde (PPA) is patterned by t-SPL. (**b**) The patterned result after t-SPL. (**c**) The final achieved nanostructure after t-SPL and laser machining. Reprinted with permission from Ref. [87]. Copyright 2022 IOP Publishing.

**Figure 15 micromachines-13-00228-f015:**
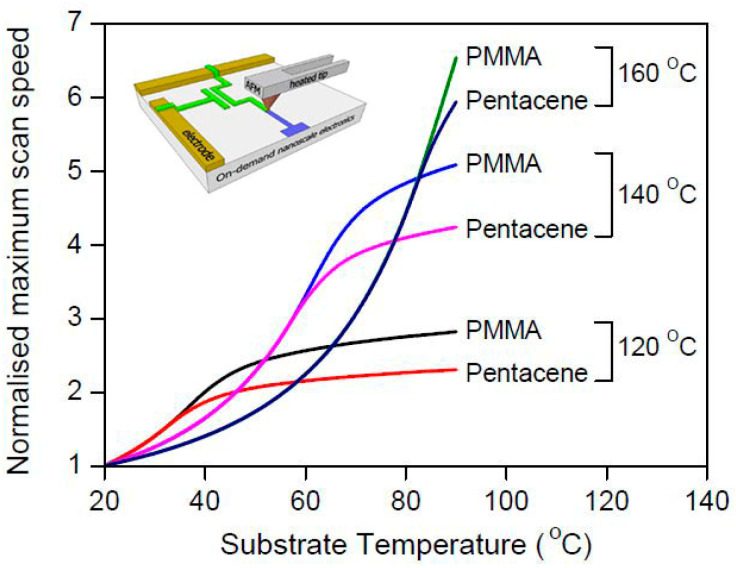
The relationship of the selective substrate temperature and maximum scanning speed. Reprinted with permission from Ref. [88]. Copyright 2022 RSC Pub.

**Figure 16 micromachines-13-00228-f016:**
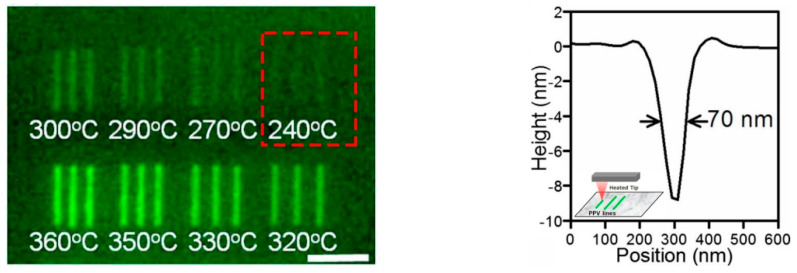
A fluorescence picture of the nanoline structure with 70 nm width under a 240 °C hot AFM tip. Reprinted with permission from Ref. [70]. Copyright 2022 AIP Publishing.

**Figure 17 micromachines-13-00228-f017:**
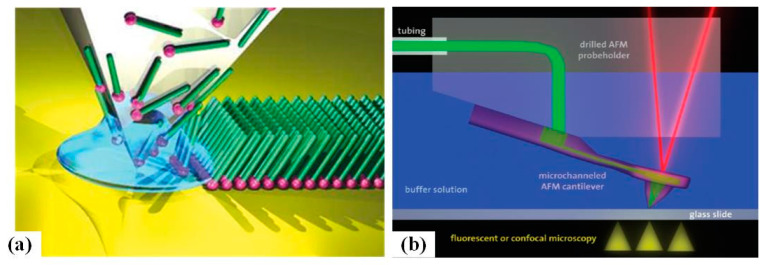
(**a**) The mechanism scheme of D-SPL. Reprinted with permission from Ref. [104]. Copyright 2022 Springer Nature (**b**) Nanofluidic delivery system. Reprinted with permission from Ref. [24]. Copyright 2022 American Chemical Society [105].

**Figure 18 micromachines-13-00228-f018:**
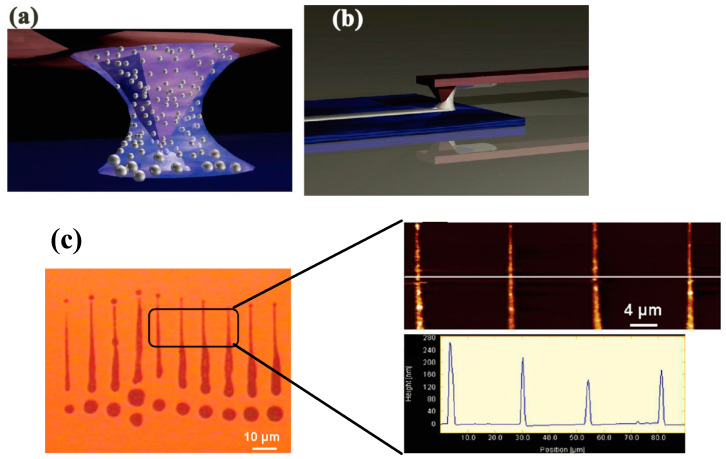
(**a**,**b**) The process mechanism schematic of depositing the Ag material nanoparticle solvent onto the SiO_2_ substrate. (**c**) The optical image and the AFM image of the Ag nanoline structures. Reprinted with permission from Ref. [107]. Copyright 2022 American Chemical Society.

**Figure 19 micromachines-13-00228-f019:**
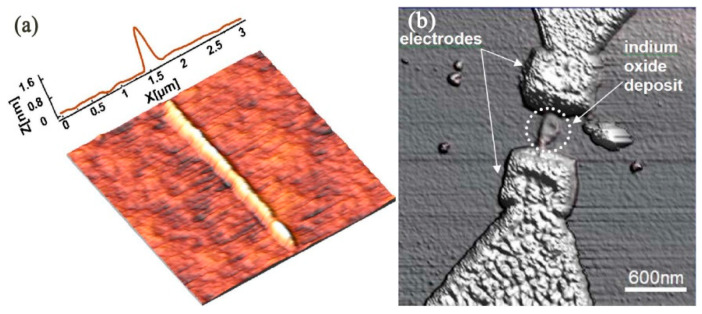
(**a**) An image of indium metal deposition on the glass substrate mixed with borosilicate. (**b**) An image of the repaired gold electrodes with a gap of around 500 nm. Reprinted with permission from Ref. [109]. Copyright 2022 AIP Publishing.

**Figure 20 micromachines-13-00228-f020:**
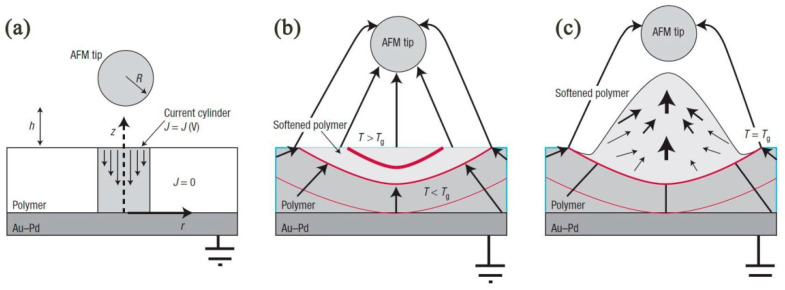
(**a**) Geometrical arrangement of AFM and polymer. Initial tip–surface distance is typically 1–5 nm. In general, the specific spatial details of the tip–surface contact profile, as well as cantilever deformation, with applied bias during writing is not well understood or documented. To a zero-order approximation, the geometrical details arising from the relative orientation of the AFM pyramidal tip with respect to the surface is ignored, and the AFM tip is approximated as a sphere of radius ~35 nm. J = J (V) is the current density, which is a function of the applied (bias) voltage. (**b**) Joule heating from amplified current flow increases temperature within the polymer film (isotherms (red solid lines) determined from time-dependent heat-transfer calculations). T > Tg defines the volume of softened viscoelastic polymer. The highly non-uniform electric field (109–1010 V m^−1^, estimated by method of images) generates a step electric field gradient (arrows). (**c**) The large non-uniform electric field gradient that surrounds the AFM tip produces an electrostatic pressure on the polarizable, softened polymer creating raised features. The schematics of the mechanism of B-SPL. Reprinted with permission from Ref. [114]. Copyright 2022 Springer Nature.

**Figure 21 micromachines-13-00228-f021:**
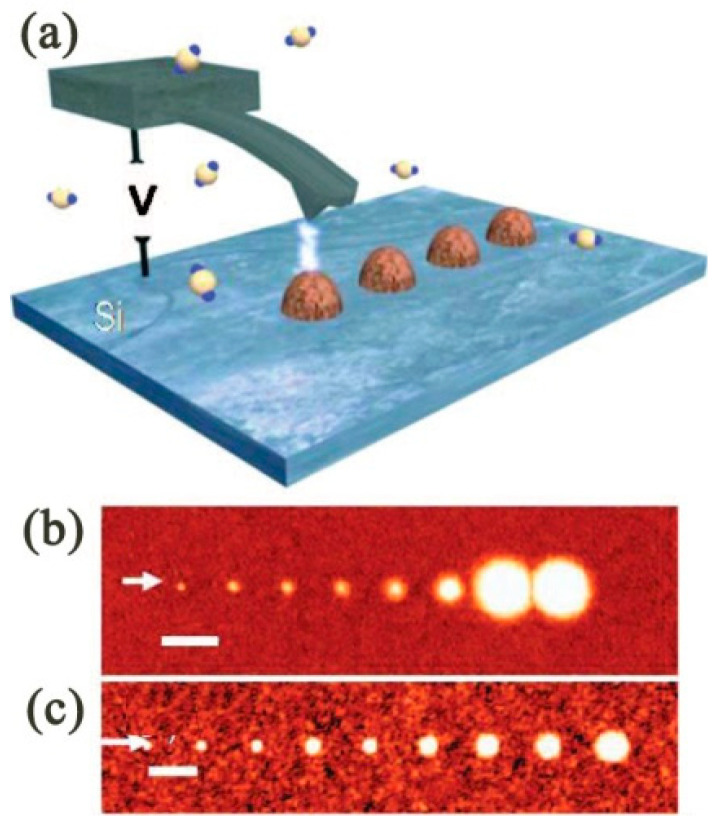
(**a**) The schematic of B-SPL for a carbon nanodot structure. (**b**) The nanodot structure under different processing times of 0.1 ms, 0.5 ms, 1 ms, 10 ms, 50 ms, 100 ms, 1s, and 2s with 21 V. (**c**) The achieved nanodot structure under different voltages of 20 V, 22 V, 24 V, 26 V, 28 V, 30 V, 34 V, and 36 V under 0.1 ms. Reprinted with permission from Ref. [126]. Copyright 2022 AIP Publishing.

**Figure 22 micromachines-13-00228-f022:**
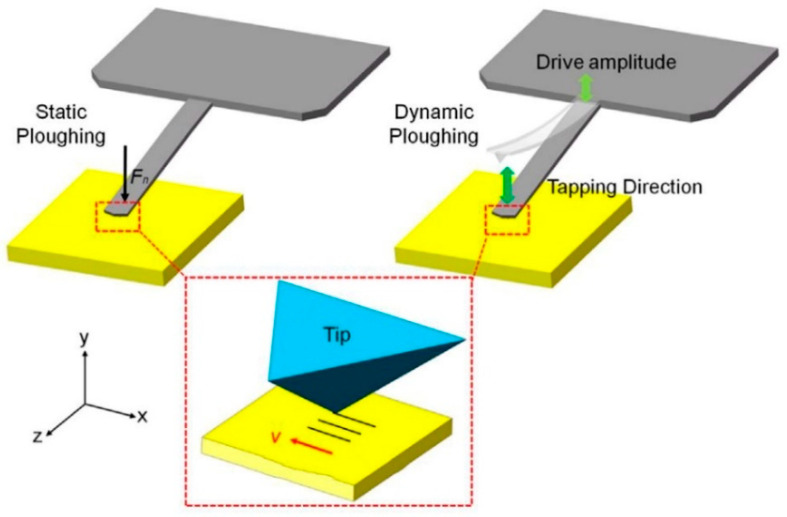
The schematic of the static ploughing lithography (left side) and dynamic ploughing lithography (right side) material removal. Reprinted with permission from Ref. [130]. Copyright 2022 Elsevier.

**Figure 23 micromachines-13-00228-f023:**
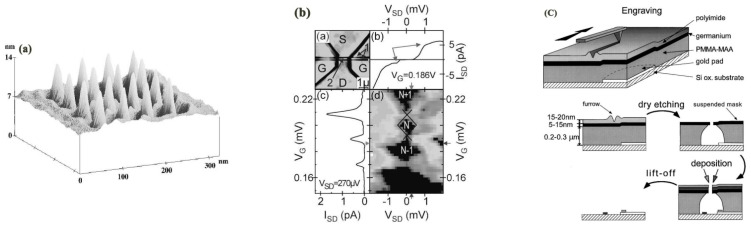
(**a**) The inverted topography of the 16 nanoscale hole array with 55 nm periodicity. (**b**) An image of the channel barrier and insulated gate. (**c**) The process of a single-electron transistor fabrication using the mixture of the mechanical approach based on AFM fabrication, dry etching, and lift off. Reprinted with permission from Ref. [134]. Copyright 2022 AIP Publishing.

**Figure 24 micromachines-13-00228-f024:**
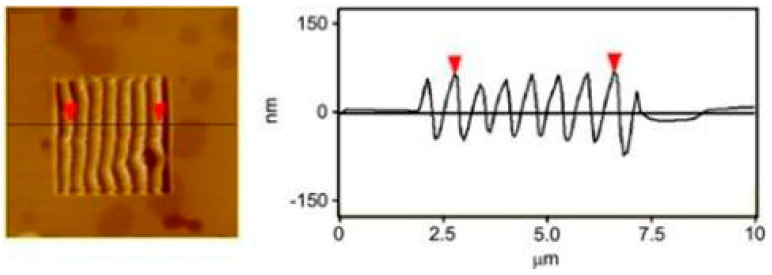
Morphology and the intersecting surface of the substrate under one time scan with a normal load of 13.6 μN. Reprinted with permission from Ref. [142]. Copyright 2022 Elsevier.

**Figure 25 micromachines-13-00228-f025:**
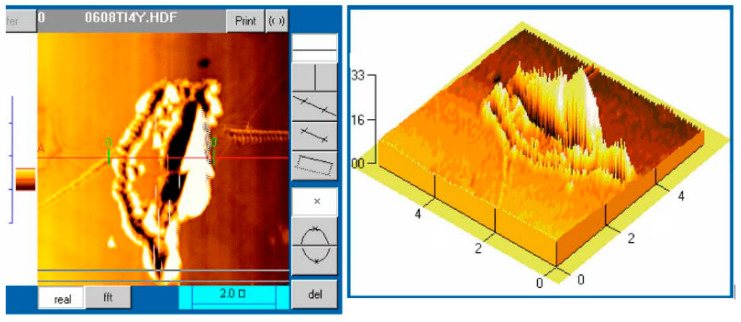
The 3D Taiwan island image. Reprinted with permission from Ref. [141]. Copyright 2022 AIP Publishing.

**Figure 26 micromachines-13-00228-f026:**
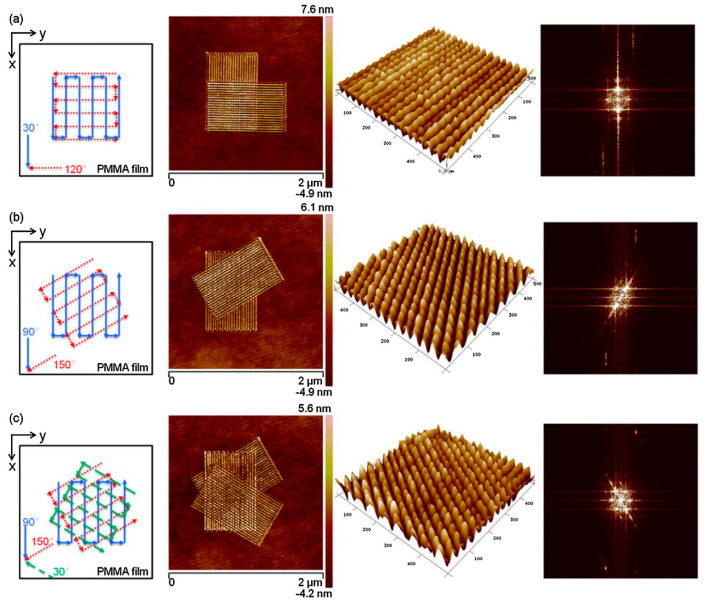
An image of various nanodots and fast Fourier transform (FFT) images. (**a**) Checkerboard nanodots from 30° and 120° machined directions. (**b**) Diamond-shaped nanodots from 90° and 150° machined directions. (**c**) Hexagonal nanodots from 30°, 90°, and 150° machined directions. Reprinted with permission from Ref. [144]. Copyright 2022 Elsevier.

**Figure 27 micromachines-13-00228-f027:**
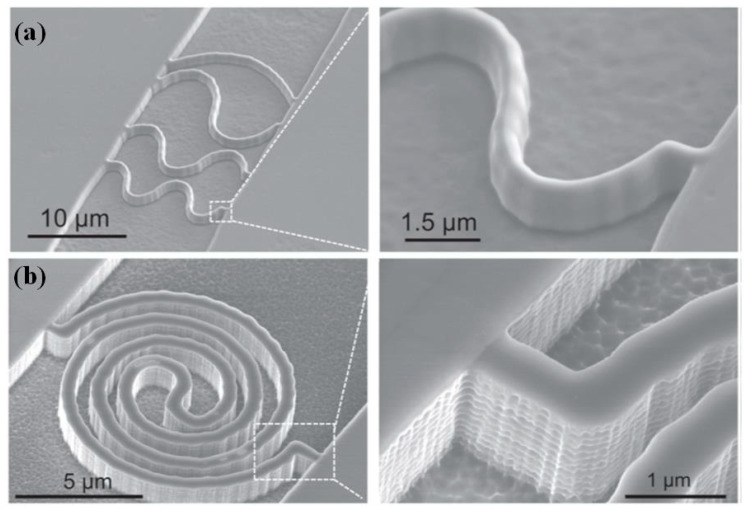
Images of nanofluidic channels: (**a**) four wavy shapes, (**b**) spiral shaped. Reprinted with permission from Ref. [155]. Copyright 2022 IOP Publishing.

**Figure 28 micromachines-13-00228-f028:**
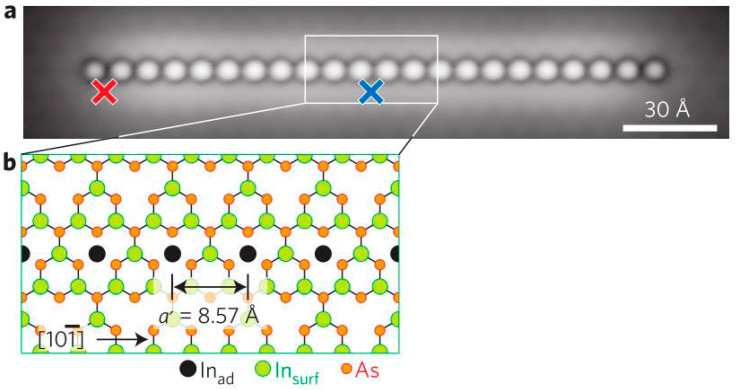
(**a**) The quantum dots constructed by a chain of 22 ionized In adatoms. (**b**) The reconstructive InAs template lattice, including black In adatoms, green In, and red As. Reprinted with permission from Ref. [159]. Copyright 2022 Springer Nature.

**Table 1 micromachines-13-00228-t001:** Comparison of current nanofabrication techniques.

Items	EBL	FIB	NIL	SPL
Principle	Physical process	Physical process	Physical and chemical process	Physical and chemical process
Machining capability	2D, 3D	2D, 3D	2D, 2.5D	2D, 3D
Resolution (nm)	5 nm [7]	5 nm [8]	10 nm [9]	See Table 2 in Section 5
Environmental conditions	Vacuum	Vacuum	Vacuum or ambient	Vacuum or ambient
Lithography speed	Slow 17 nm/min [10], 40 nm/min [11], and 58 nm/min [12] against different resists	Slow 0.05 μm^3^/s in FIB deposition [13]	Fast 15 wafers/h per imprint station [14]	See Table 2 in Section 5
Cost	Higher start-up cost	Higher start-up cost	Low	Low
Contaminated patterns	Yes	Yes	No	No

**Table 2 micromachines-13-00228-t002:** Comparison of machining capabilities of the major SPL nanolithography approaches.

Items	Close-to-Atomic Scale SPL	T/tc-SPL	O-SPL	M-SPL	D-SPL	B-SPL
Resolution	Atomic scale [3]	10 nm [85]	4 nm [60]	10 nm [165]	10 nm [74]	10 nm [166]
Throughput	-	~10^4^ μm^2^ h^−1^ [167]	~10^2^ μm^2^ h^−1^ [66]	~2.4 × 10^3^ μm^2^ h^−1^ [112]	~10^4^ μm^2^ h^−1^ [98]	~10 μm^2^ h^−1^ [168]
Machining capability	-	2D, 3D	2D, 3D	2D, 3D	2D	2D
Machinable materials	Molecular, atoms, electrons	PMMA, PC PPV film, copolymer film	Metal, semiconductors, graphene, polymer	Polymer, metal, ceramics and semiconductors, graphene	Transporting organic molecules, polymers, DNA, proteins and metal ions	Graphene, metal, semiconductors, Si, polymer
Environmental conditions	Vacuum	Liquid, Air	20%–80% relative humidity	Air	34% relative humidity	High electric fields
Processing speed	Super slow 80 nm/s [169]	Super fast 20 mm/s [170]	Moderate 10 μm/s [74]	Fast 50 μm/s [2]	Slow 2 μm/s [171]	Slow 0.1 μm/s [172]
Control	Difficult	Good	Excellent	Excellent	Complicated	Difficult
Principle	Physico-chemistry process	Physico-chemistry process	Chemical process	Physical process	Chemical process	Physico-chemistry process
Tip wear	Negligible	Not serious	Negligible	Serious	Negligible	Negligible
Advantages	Atomic-scale precision	Super fast	Robust oxide formation	Easy to implement and various substrates materials	Very suitable for biological materials	Negligible probe wear
Disadvantages	Extreme slow	Requires heated probes	Requires oxidizability of the workpiece	Probe wear and burr formation	Requires ink	Requires extra electric circuits to control current

## Data Availability

No new data were created during this study. Pre-existing data underpinning this publication were obtained from published papers.

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
