# Peer review of "Scanning Probe Lithography: State-of-the-Art and Future Perspectives"

_micromachines, 2022, doi:10.3390/mi13020228_

Round 1
Reviewer 1 Report
Please find attached.

Reviewer 2 Report
Review of
Scanning probe lithography: state-of-the-art and future perspectives
This paper presents a review of the state of the art of scanning probe based lithography techniques and prospects for these techniques to improve to the point that they would be industrially relevant lithography tools.
To their credit, the authors cover a wide range of techniques and their applications and potential applications.
However the paper as written has weaknesses.
While there are many SPL techniques and not all of them need to be covered, I believe a major gap is that there is no mention of Hydrogen Depassivation Lithography as developed by Lyding[1] and used extensively by Simmons[2] and others in the pursuit of quantum computers and other quantum devices by the placement of dopant atoms with near atomic precision in single buried Si (100) planes. The applications of this technique has also demonstrated a number of other pattern transfer methods including the selective deposition of TiO2 creating etch masks[3], the creation of atomically precise dangling bonds on Si surfaces as quantum dots for quantum and other device uses[4] and the use of halogens[5] and molecules[6] as alternative resists.
While they do acknowledge the single largest barrier to commercial use, patterning throughput, they only briefly mention attempts at parallelization where there have been some discussion in the literature and the issues with tip changes during patterning along with reliability and automation that will be required for improving throughput. I would have liked to see more discussion of these issues that are relatively common to all SPL technologies.
Some of the language and characterizations of patterning speed are misleading in my estimation. For instance, in Table 1 the Lithography Speed for SPL is listed as “Fast”. By what measure I would ask? Being quite familiar with E-Beam Lithography and SPL, I have an issue with characterizing SPL as fast and EBL as slow. In this and several other general characterizations of lithographic techniques throughout the manuscript, the authors have failed to explain how they are quantifying their metrics.
In Table 1, I also have an issue of the resolutions listed for NIL and SPL. Much higher resolution is clearly available for NIL[7] and the authors mention several SPL techniques with near atomic resolution which is smaller than 10nm but since they list <5nm for EBL and FIB neither of which have near atomic resolution this is an inconsistent description of a resolution comparison.
There are several other inconsistencies in the article. I mention a few below but there are others which I believe are a result of the authors not identifying how they are quantifying their metrics:
Lines 43 – 44: FIB removal rate is slow, how is removal rate relative to e-beam litho which exposes a polymer resist?
Lines 50-51 – the problems listed with the molds are not relevant to the leading NIL technology from Canon Nanotech.
Line 375 – extremely rapid only applies to simple small patterns. Even for research any serial write approach will have throughput challenges
I do not want to sound overly critical. The authors have covered a lot of ground in describing a large number of SPL techniques and provide a lot of valuable information. I would like to see this published, but believe it would be a much better article if the authors would better explain how they are quantifying their metrics and to cover Hydrogen Depassivation Lithography which is their most significant gap in covering SPL techniques.
[1] Hersam, M. C., Guisinger, N. P., & Lyding, J. W. (2000). Silicon-based molecular nanotechnology. Nanotechnology, 11(2), 70–76. https://doi.org/10.1088/0957-4484/11/2/306
[2] Hill, C. D., Peretz, E., Hile, S. J., House, M. G., Fuechsle, M., Rogge, S., Simmons, M. Y., & Hollenberg, L. C. L. (2015). A surface code quantum computer in silicon. Science Advances, 1(9), e1500707–e1500707. https://doi.org/10.1126/sciadv.1500707
[3] Ballard, J. B., Owen, et.al. (2014). Pattern transfer of hydrogen depassivation lithography patterns into silicon with atomically traceable placement and size control. Journal of Vacuum Science & Technology B, Nanotechnology and Microelectronics: Materials, Processing, Measurement, and Phenomena, 32(4), 041804. https://doi.org/10.1116/1.4890484
[4] Achal, R., et.al.. (2018). Lithography for Robust and Editable Atomic-scale Silicon Devices and Memories. Nature Communications, 2018. https://doi.org/10.1038/s41467-018-05171-y
[5] Dwyer, K. J., Dreyer, M., & Butera, R. E. (2019). STM-Induced Desorption and Lithographic Patterning of Cl–Si(100)-(2 × 1). The Journal of Physical Chemistry A, 123(50), 10793–10803. https://doi.org/10.1021/acs.jpca.9b07127
[6] Kruse, P., & Wolkow, R. a. (2002). “Gentle lithography” with benzene on Si(100). Applied Physics Letters, 81(23), 4422. https://doi.org/10.1063/1.1526459
[7] http://cnt.canon.com/technology/imprint-resolution-and-pattern-transfer/
